# Decursin and Decursinol Angelate Suppress Adipogenesis through Activation of β-catenin Signaling Pathway in Human Visceral Adipose-Derived Stem Cells

**DOI:** 10.3390/nu12010013

**Published:** 2019-12-19

**Authors:** In Sil Park, Boyun Kim, Youngjin Han, Hee Yang, Untack Cho, Se Ik Kim, Jong Hun Kim, Jung Han Yoon Park, Ki Won Lee, Yong Sang Song

**Affiliations:** 1Department of Agricultural Biotechnology, Seoul National University, Seoul 08826, Korea; insil@snu.ac.kr (I.S.P.); yhee6106@snu.ac.kr (H.Y.); 2Cancer Research Institute, Seoul National University College of Medicine, Seoul 03080, Korea; boyunism@gmail.com (B.K.); youngjin.han@snu.ac.kr (Y.H.); 3Department of Anesthesiology, McGovern Medical School, The University of Texas Health Science Center, Houston, TX 77030, USA; 4Biomodulation, Department of Agricultural Biotechnology, Seoul National University, Seoul 08826, Korea; 5Interdisciplinary Program in Cancer Biology, Seoul National University, Seoul 03080, Korea; jut2805@snu.ac.kr; 6Department of Obstetrics and Gynecology, Seoul National University College of Medicine, Seoul 03080, Korea; seikky@naver.com; 7Department of Food Science and Biotechnology, Sungshin Women’s University, Seoul 01133, Korea; killrose@hotmail.com; 8Research Institute of Agriculture and Life Sciences, Seoul National University, Seoul 08826, Korea; junghanyoonpark@gmail.com; 9Advanced Institutes of Convergence Technology, Seoul National University, Suwon 16229, Korea

**Keywords:** *Angelica gigas* Nakai, decursin, decursinol angelate, visceral obesity, adipose-derived stem cells, adipogenesis, β-catenin

## Abstract

Visceral adiposity is closely associated with metabolic disorders and cardiovascular diseases. *Angelica gigas* Nakai (AGN) has been reported to possess anti-obesity effects and higher amounts of coumarin compounds are present in AGN. However, the active compounds suppressing adipogenesis in AGN and mechanisms of action have not been investigated in adipose-derived stem cells (ASCs) isolated from visceral adipose tissue (VAT). Among four coumarin compounds of AGN, decursin (D) and decursinol angelate (DA) significantly inhibited adipocyte differentiation from ASCs. D and DA downregulated CCAAT/enhancer binding protein α (C/EBPα), peroxisome proliferator-activated receptor γ (PPARγ), adipocyte fatty acid binding protein (aP2), fatty acid synthase (FAS), and acetyl-CoA carboxylase (ACC) at both mRNA and protein levels. Next, treatment with adipogenic differentiation medium (ADM) on ASCs downregulated β-catenin expression at protein level, while addition of D and DA could restore protein expression and nuclear translocation of β-catenin suppressed by ADM. D and DA treatment on ADM treated ASCs increased inhibitory phosphorylation of Glycogen synthase kinase (GSK)-3β, thereby preventing β-catenin from degradation. Additionally, si-β-catenin transfection significantly upregulated protein expression of C/EBPα and PPARγ, alleviating the anti-adipogenic effect of D and DA on ADM treated ASCs. Overall, D and DA, active compounds from AGN, suppressed adipogenesis through activation of β-catenin signaling pathway in ASCs derived from human VAT, possibly using as natural anti-visceral adiposity agents.

## 1. Introduction

Obesity is excessive fat deposition and has a profound impact on quality of life. Specifically, visceral obesity characterized by high visceral adipose tissue (VAT) distribution in intra-abdominal, associated with a high risk of metabolic disorders including type 2 diabetes, hypertension, and atherosclerosis [1,2]. Obesity is induced by adipogenesis impairments of adipocytes in adipose tissue increasing lipid accumulation. Adipogenesis is a complex process which includes increase of adipocyte differentiation from adipose-derived stem cells (ASCs) and intracellular lipid accumulation [3]. In our previous study, we showed differential gene expression pattern of ASCs between human VAT and subcutaneous adipose tissue (SAT). VAT-derived ASCs showed higher expression of gene clusters involved in lipid biosynthesis than SAT-derived ASCs, suggesting that VAT-derived ASCs may have higher adipogenic potential [4]. Therefore, inhibition of adipogenesis of VAT-derived ASCs could be an ideal therapeutic strategy for treating visceral obesity.

Adipocyte differentiation from ASCs is regulated by key adipogenic transcription factors, including members of the CCAAT/enhancer binding protein (C/EBP) family and peroxisome proliferator-activated receptor γ (PPAR-γ) [5,6]. These transcription factors are crucial for expression of adipocyte-specific genes, such as adipocyte fatty acid binding protein (aP2), fatty acid synthase (FAS) and acetyl-CoA carboxylase (ACC) [7]. Moreover, increasing evidences suggest that β-catenin inhibit adipogenesis by blocking interaction between the TCF/LEF-binding domain of β-catenin and the catenin-binding domain of C/EBPα/and PPARγ. In addition, in the nucleus, β-catenin upregulates its target gene cyclin D1 (CCND1), which downregulates key adipogenic transcription factors, C/EBPα and PPARγ [8,9,10]. Therefore, upregulation of β-catenin pathway could be considered as effective target signaling in inhibiting adipogenesis.

*Angelica gigas* Nakai (AGN) is also known as ‘Korean dang-gui’. Higher amounts of coumarin compounds such as decursin (D), decursinol angelate (DA), nodakenin (ND) and 7-dimethyl suberosin (DS) are present in AGN than in other Angelica species such as *Angelica sinesis* (Chinese dang-gui) and *Aneglica acutiloba* (Japanese dang-gui) [11]. However, anti-adipogenic effects of active compounds present at high level in AGN has not been investigated in VAT-derived ASCs though AGN has been traditionally used as medication for treating metabolic disorders. Also, the underlying mechanism of action remains to be determined.

Herein, we aimed to investigate anti-adipogenic effect of the active compounds present in AGN and their mechanisms of action using ASCs isolated from human VAT. Our findings show that D and DA, the active compounds of AGN, suppress adipogenesis of ASCs through activation of β-catenin signaling pathway.

## 2. Materials and Methods

### 2.1. Reagents

D and DA were purchased from Biopurify Phytochemicals Ltd. (Chendu, China) with 98% purity. The chemical structure of D and DA is shown in Figure 1b. ND and DS were purchased from Chem Faces (Hubei, China) and the purity is 98%. The anti-CD31, CD45 microbeads and magnetic cell sorting system (MACS) separation buffer were purchased from Miltenyl Biotec (Bergisch Galdbach, Germany). MesenPRO RS medium, glutaMAX and insulin were purchased from Gibco (Waltham, MA, USA). Collagenase type IA, 3-isobutyl-1-methylzanthine (IBMX), indomethacin, dexamethasone and Oil-Red O were purchased from Sigma-Aldrich (St. Louis, MO, USA). Trizol and reverse transcriptase were purchased from TAKARA (Kusatsu, Japan). SYBR supermix was purchased from Bio-Rad Laboratories (Hercules, CA, USA). NE-PER nuclear and cytoplasmic extraction reagents and BCA protein assay kit were purchased from Thermo Fisher Scientific (Waltham, MA, USA). Antibodies against C/EBPα (#2295), PPARγ (#2430), FAS (3180S), ACC (3662S), p-GSK-3β (S9) (#9323), and total GSK-3β (#9315) were purchased from Cell signaling (Danvers, MA, USA). Antibodies against aP2 (sc-365236), ERK1/2 (sc-514302) and p-ERK1/2 (sc-7383) were purchased from Santa Cruz Biotechnology (Dallas, TX, USA). Antibodies against active β-catenin (05–665) was purchased from EMD Millipore (Burlington, MA, USA). Triglyceride quantification assay kit (ab65336), antibodies against β-catenin (ab32572) and goat anti-rabbit IgG (ab150077) were purchased from Abcam (Cambridge, MA, USA). Antibody against GAPDH (LF-PA0202) was purchased from Ab Frontier (Seoul, Korea). Four-well chamber slides with removable wells were purchased from NuncTM LabTek II 4-well chamber slideTM. Amplex Red cholesterol assay kit (A12216), Opti-MEM, normal goat serum control, ProLongTM Glass Antifade Mountant and lipofectamineTM 2000 transfection reagent were purchased from Invitrogen (Waltham, MA, USA).

### 2.2. Human ASCs and Culture

VAT (intra-abdominal) samples were obtained from retroperitoneal fat tissue of human donors (*n* = 5) receiving gynecologic surgery. Clinical information regarding these patients is shown in Table 1. The procedure was approved by Seoul National University Hospital Institutional Review Board (SNU-1003-009-311). We have followed the provisions of the Declaration of Helsinki and obtained informed consent from the human donors for this research. ASCs were isolated as described previously [4]. Briefly, VAT was washed with sterile PBS and blood vessels were removed. Processed VAT was then dissociated with collagenase type IA diluted in 0.25 mg/mL PBS for 1 h at 37 °C was centrifuged at 500× *g* for 4 min. After centrifugation, stromal vascular fraction (SVF) pellet was collected. The SVF was filtered using MACS through negative selection of CD31 (endothelial cell marker) and CD45 (hematopoietic stem cell marker). The CD31 negative and CD45 negative SVF was plated onto 100 mm culture dish in MesenPRO RS medium containing 1% glutaMAX and 1% penicillin-streptomycin. After 3 days, non-adherent cells were removed by washing with PBS. ASCs were obtained by expansion of the adherent cells and they were passaged at least two times prior to experimental use. ASCs at passages 3–6 were used for experiments.

### 2.3. Adipocyte Differentiation

ASCs were seeded onto 24-well plates at 0.015 × 10^6^ cells/well and cultured in MesenPRO RS medium. When the cells reached confluence, they were incubated with adipogenic differentiation med (ADM) which is DMEM F-12 supplemented with 10% FBS, 1% PS, 10 μg/mL insulin, 0.5 mM IBMX, 50 μM indomethacin, and 1 μM dexamethasone. The medium was changed every second day and the cells were cultured for 14 days. Differentiation of ASCs into adipogenic lineage was examined by Oil Red O (ORO) staining. Briefly, differentiated adipocytes were washed with PBS, and then fixed in 4% PFA for 30 min. The fixed cells were then washed with deionized water, and 60% saturated ORO staining was carried out for 1 h. For ORO quantification, isopropanol was added to each well. Light absorbance was measured at 495 nm.

### 2.4. Cholesterol and Triglyceride Content

To analyze the intracellular content of cholesterol, the cells were washed with PBS, the neutral lipids were extracted by 1 mL of mixture of chloroform and methanol (3:2) for 30 min. The lipids were dried in the fume hood. Cholesterol content was determined using Amplex red cholesterol assay kit according to the manufacturer’s protocol. Intracellular triglyceride was measured in cell lysates by a colorimetric triglyceride quantification assay kit according to the manufacturer’s instruction.

### 2.5. qRT-PCR

Total RNA was extracted from ASCs using Trizol reagent and RNA concentration was determined by Nanodrop (Nano Drop 2000, Thermo Scientific). The cDNA was obtained by reverse transcription containing 1 μg of total RNA, oligo (dT), and reverse transcription premix. The qRT-PCR reactions were performed with the SYBR green PCR system (CFX96, Bio-Rad) in a thermal cycler (C1000, Bio-Rad). The cycling conditions were as follows: 95°C for 10 min; followed by 40 cycles involving denaturing at 95 °C for 5 s, annealing at 60 °C for 15 s, and extension at 72 °C for 10 s. Expression of mRNAs was normalized by the mRNA levels of GAPDH. The following forward and reverse primer sequences were used: C/EBPα, 5′-GCAAACTCACCGCTCCAATG-3′,5′-CTTCTCTCATGGGGGTCTGC-3′; PPARγ:5′-AGGTCAGCGGACTCTGGATTC-3′,5′-AGTGGGGATGTCTCATAATG-3′; aP2, 5′-ATGGGGGTGTCCTGGTACAT-3′, 5′-ACGTCCCTTGGCTTATGCTC-3′; LPL: 5′-CGAGCGCTCCATTCATCTCT-3′, 5′-CCAGATTGTTGCAGCGGTTC-3′; FAS, 5′-AAGGACCTGTCTAGGTTTGATGC-3′, 5′-TGGCTTCATAGGTGACTTCCA-3′, ACC, 5′-CTTGAGGGCTAGGTCTTTCTGG-3′, 5′-CTGGTTCAGCTCCAGAGGTT-3′, SREBP-1c, 5′-GCCCCTGTAACGACCACTG-3′, 5′-CAGCGAGTCTGCCTTGATG-3′, GAPDH, 5′-GAGTCAACGGATTTGGTCGT-3′,5′-TTGATTTTGGAGGGATCTCG-3′.

### 2.6. Western Blot

ASCs were lysed with lysis buffer supplemented with 1% triton X-100, EDTA free protease inhibitor cocktail, Na3VO4, phenyl methyl sulfonyl fluoride, and sodium deoxycholate. Nuclear and cytoplasmic fraction were prepared according to the manufacturer procedure of NE-PER nuclear and cytoplasmic extraction reagents. Protein concentrations were measured using BCA protein assay kit. Protein samples (5 μg per well) in each sample were separated by 9% SDS-PAGE and transferred to a nitrocellulose membrane. The membrane was blocked with 5% skim milk in Tris-buffered saline containing 0.1% Tween 20 (TBS-T) and incubated with primary antibodies (1:1000 dilution) overnight at 4 °C. Immunoblot analyses were performed using the antibodies. The membrane was incubated with peroxidase-conjugated secondary antibody (1:5000 dilution). Target proteins were detected using a western blot detection kit.

### 2.7. Immunocytochemistry

ASCs cells were seeded on four-well chamber slides at a density of 0.01 × 10^6^ cells per well. After 3 days incubation, ASCs were cultured in ADM with or without D and DA treatment for additional 2 days. The cells were subsequently washed and fixed with 4% PFA solution for 20 min. Cells were permeabilized with 0.25% Triton X-100 for 20 min and blocked with 10% normal goat serum control in PBS/T for 1 h. The cells were incubated with β-catenin antibody overnight at 4 °C and then washed with 1% goat serum in PBS/T three times. Cells were then incubated with Alexa Fluor 488-conjugated IgG secondary antibody in 10% goat serum with PBS/T for 1 h and washed with 1% goat serum in PBS/T three times at dark. A drop of mounting solution with DAPI was added to each slide. After mounting, the fluorescence signal was captured under confocal microscopy (LSM800, Zeiss). To quantify nuclear β-catenin signals per cell, the number of cells with β-catenin signal in nucleus (*n* = 90–100) in each group of samples in two independent experiments was measured.

### 2.8. siRNA Transfection

ASCs were seeded in 6-well plates at a density of 0.04 × 10^6^ cells per well. The transfection was conducted in Opti-MEM and performed as described in instruction of lipofectamineTM 2000 transfection reagent. ASCs were transfected with 50 nM of β-catenin or negative control siRNAs. siRNAs were synthesized by Genolution (Korea). The following forward and reverse primer sequence were used: 5′-GUGCUAUCUGUCUGCUCUADTDT-3′,5′-UAGAGCAGACAGAUAGCACDTDT -3′. After 8 h of transfection, the medium was changed and the cells were incubated for another 2 days to reach at confluence. The transfected cells were cultured in ADM with or without D and DA treatment.

### 2.9. Statistical Analysis

Values are expressed as mean ± SD of three independent experiments. For multiple comparisons, analysis of variance (ANOVA) was used followed by Tuckey’s test. Statistical analysis was performed with SPSS (version 23) program. Differences are regarded significant if the value of *p* < 0.05.

## 3. Results

### 3.1. D and DA from AGN Inhibit Adipogenesis in ASCs

To evaluate the anti-adipogenic effect of active compounds from AGN, ASCs were differentiated with ADM in the presence of coumarin compounds (D, DA, ND, and DS) from AGN at 40 μM for 14 days (Figure 1a). Cytotoxicity of the coumarin compounds was assessed in ASCs by MTT assay. None of the compounds that we tested were showing cytotoxic effect on ASCs at 40 μM after 48 h incubation (Figure 1c). The intracellular lipid content was assessed by Oil Red O (ORO) staining 14 days after adipogenic induction with ADM. Quantitative analysis of the ORO staining revealed that ADM treated group exhibited 4.6-fold higher intensity of ORO staining than the undifferentiated group. However, D (40 μM) and DA (40 μM) treated groups showed 1.91-fold and 1.90-fold less intensity of ORO-staining respectively than ADM-only treated group, whereas ORO staining intensity was not affected by addition of ND and DS on ADM-treated ASCs. Consistent with above results, less adipocytes with ORO-stained lipid droplets were present in D and DA treated group than ADM-only treated group (Figure 1d,e). These data indicate that among four coumarin compounds from AGN, D, and DA are the most effective compounds at suppressing adipogenesis. To determine the optimal concentration of D and DA with minimal cytotoxicity on ASCs and anti-adipogenic effect, D and DA were treated with increasing concentrations (0, 5, 10, 20, and 40 μM) and we found that D and DA did not exhibit cytotoxicity after 48 h incubation (Figure 1f,h). D (40 μM) and DA (40 μM) significantly suppressed lipid accumulation, inhibiting induction of adipogenesis (Figure 1g,i). To further investigate the types of lipid changed by D and DA, triglyceride and cholesterol concentration were quantified after adipogenic induction with ADM and addition of D and DA on ADM treated ASCs. ADM treatment significantly increased triglyceride contents in ASCs. While, the addition of D (40 μM) and DA (40 μM) on ADM treated ASCs significantly decreased triglyceride concentration (Appendix A). On the other hand, cholesterol concentration showed no statistically significant difference among ADM, D (40 μM), and DA (40 μM) groups (Appendix A). Therefore, these results indicate that among major coumarin compounds of AGN, D and DA showed anti-adipogenic effect on visceral ASCs.

### 3.2. D and DA Downregulate Expression of Adipogenic and Lipogenic Markers in ASCs

Next, we conducted further experiments with D and DA concentration fixed at 40 μM to find molecular signaling responsible for anti-adipogenic effect. To determine whether D and DA affect the expression of adipogenic transcription factors, we performed qRT-PCR 3 days after adipogenic induction with ADM treatment. The mRNA levels of adipogenic genes, such as C/EBPα, PPARγ, aP2, and lipoprotein lipase (LPL) were significantly increased by ADM exposure. However, upregulated the adipogenic genes with ADM treatment were significantly suppressed by addition of D and DA to ADM (Figure 2a–d). To examine the effect of D and DA on protein levels of the adipogenic markers, cells were harvested 4 days after ADM treatment. Protein expression of C/EBPα, PPARγ, and aP2 was markedly upregulated by ADM treatment, while the increases of adipogenic marker expression was reversed by the addition of D and DA to ADM (Figure 2e–h). Adipocyte differentiation can activate key lipogenic enzymes such as fatty acid synthase (FAS) and acetyl CoA carboxylase (ACC). FAS promotes the synthesis and storage of triglyceride in adipocytes and ACC converts malonyl-CoA into palmitate [7]. We examined the effects of D and DA on expression levels of lipogenic markers 12 days after adipogenic induction with ADM treatment. The mRNA expression of FAS, ACC, and sterol regulatory element-binding transcription factor (SREBP)-1c significantly decreased by D and DA treatment as compared to the cells with ADM exposure (Figure 2i–k). Consistently, the protein levels of FAS and ACC were increased by ADM treatment, while this increase of lipogenic marker expression was reduced in D and DA treated cells after ADM treatment (Figure 2i–n). These results indicate that D and DA inhibited differentiation of ASCs into adipocytes and lipid accumulation in parallel with downregulation of mRNA and protein expressions of the adipogenic and lipogenic markers.

### 3.3. D and DA Suppress Adipogenesis through Upregulation and Nuclear Translocation of β-Catenin in ASCs

We explored the mechanism of anti-adipogenesis by D and DA. Recently, reduction in β-catenin content in ASCs has been shown to facilitate adipocyte differentiation [12]. So, we examined the effect of D and DA on β-catenin expression in treatment of ADM to ASCs. To address this question, we observed changes in β-catenin protein expression of ASCs in different experimental conditions. ADM-mediated adipogenic differentiation of ASCs resulted in a downregulation of β-catenin expression at protein level. Whereas, β-catenin expression was significantly higher in D and DA-treated group than in ADM only treated group. Expression of active β-catenin (non-phosphorylated β-catenin) was also upregulated by the addition of D and DA in the presence of ADM (Figure 3a). Also, we found that in ADM treated groups β-catenin content in nucleus was elevated with the addition of D and DA, while there was no significant change of β-catenin level in cytosol when D and DA were treated (Figure 3b–d). Consistent with this, treatment of D and DA promoted the translocation of β-catenin from the cytoplasm into the nucleus and this was confirmed by immunocytochemistry (Figure 3e). Moreover, treatment of D and DA resulted in a marked increase in the proportion of cells positive for β-catenin in the nucleus significantly (Figure 3f). Taken together, D and DA increased expression and the nuclear translocation of β-catenin in ADM treated ASCs, resulting in suppression of adipogenesis.

### 3.4. Upregulation of β-catenin by D and DA is Mediated via Inhibitory Phosphorylation of GSK-3β in ASCs

To investigate the upstream regulator of β-catenin during adipogenesis of ASCs, ERK 1/2 expressions were analyzed. Treatment with ADM on ASCs increased inhibitory phosphorylation of GSK-3β at S9 residue [13]. In ASCs cultured in ADM, inhibitory phosphorylation of GSK-3β at S9 was increased in D and DA treated group (Figure 4a,b). Furthermore, ratio of phosphorylated/total ERK 1/2 (T202/Y204) was increased by D and DA treatment in ADM-cultured ASCs (Figure 4c,d). These findings suggest that treatment of D and DA may inhibit GSK-3β through activation of ERK as the upstream regulator.

### 3.5. Downregulation of C/EBPα and PPARγ by D and DA is Mediated via Activation of β-Catenin

To further confirm whether inhibition of adipogenesis by D and DA is mediated via activation of β-catenin, we conducted knockdown of β-catenin using si-RNA in ASCs. Transfection with si-β-catenin significantly downregulated β-catenin expression. Transfection with si-β-catenin into ASCs suppressed β-catenin expression induced by treatment with D and DA. In addition, C/EBPα and PPARγ have been reported as the downstream targets of β-catenin signaling in 3T3-L1 murine preadipocytes [14,15]. Therefore, we examined the expression of C/EBPα and PPARγ were effectively reversed by β-catenin siRNA compared to control siRNA in ASCs. The data showed that transfection with si-β-catenin into ASCs increased C/EBPα and PPARγ expression, suppressed by treatment with D and DA (Figure 5a). Lastly, knock down of β-catenin increased lipid accumulation inhibited by D and DA treatment (Figure 5b,c). Taken together, activation of β-catenin plays a critical role in anti-adipogenic effect of D and DA. These results suggest that D and DA treatment could facilitate the anti-adipogenenic effect by downregulating C/EBPα and PPARγ through activation and nuclear translocation of β-catenin.

## 4. Discussion

We demonstrate for the first time anti-adipogenic effect of D and DA, active compounds from AGN, through activation of β-catenin signaling pathway in human visceral ASCs. AGN has been shown to reduce body weight significantly in the high-fat diet (HFD) mouse model [16]. Although anti-obesity effect of AGN has been previously shown in the mouse model, active compounds present in AGN and their role in adipogenesis is yet to be determined, particularly in human ASCs from visceral depot.

The formation of adipose tissue from adipocytes in our body is important in maintaining metabolism. However, excessive accumulation of VAT is associated with incidence of chronic non-communicable diseases such as cardiovascular disease and cancer and contributes to induction of inflammation, hypoxia, fibrosis, and necrosis [17,18,19]. So, we focused on prevention of the excessive adipogenesis of VAT. Expansion of VAT by increasing the differentiation of adipocytes from ASCs [20]. Therefore, inhibition of adipogenesis of VAT-derived ASCs could be an ideal therapeutic strategy for treating visceral obesity and VAT-derived ASCs may be an optimal in vitro model for studying adipogenesis. Waist-to-hip ratio (WHR) is recognized as a strong predictor of visceral obesity [21]. Based on WHR, female individuals could be categorized into visceral obesity (WHR ≥ 0.85) and normal (WHR < 0.85) groups. To examine the effect of AGN on ASCs from human VAT, we used the ASCs obtained from 5 women with visceral obesity (Table 1).

ASCs used in this study had a different adipogenic potential. ADM-treated ASC #20 showed 1.32-fold higher ORO-staining intensity than the undifferentiated parental ASCs (*p* > 0.05). Interestingly, the other ASCs showed significantly higher ORO-staining intensity than ASCs #20 (Appendix A). Therefore, we decided to exclude ASCs #20 in further analysis. This might be due to the individual variations in the biological nature of ASCs. Thus, innate factors of ASCs may affect the adipogenesis.

D and DA inhibited differentiation of ASCs into adipocytes in our study, verified by ORO-staining of lipid contents in adipocytes. However, there have been contradictory results about the effect of AGN on adipogenesis. Previous study showed that AGN promoted adipocyte differentiation through activation of insulin signaling pathway in 3T3-L1 murine preadipocytes [22]. This could result from disparities of genomic constituents involved in energy metabolism between rodent and human.

D and DA increased nuclear translocation of β-catenin, together with inhibition of adipogenic differentiation of ASCs. Recent studies highlighted the involvement of β-catenin signaling during adipogenesis [23,24]. These studies support our results. However, in prostate cancer, D inhibited β-catenin signaling pathway through the degradation of cytoplasmic β-catenin, inhibiting the growth of PC3 prostate cancer cells [25]. Thus, the effect of D on β-catenin signaling pathway could be cell-type dependent.

In the absence of Wnt signal, β-catenin phosphorylated by GSK-3β undergoes proteasomal degradation. Activation of Wnt signaling leads to inhibitory phosphorylation of GSK-3β at S9 followed by increased nuclear translocation of non-phosphorylated β-catenin [26]. Consistent with previous studies, D and DA treatment enhanced inhibitory phosphorylation of GSK-3β (S9) followed by nuclear translocation of β-catenin in ASCs. Thus, accumulation of nuclear β-catenin can suppress adipogenesis via downregulation of C/EBPα and PPARγ, key transcription factors associated with adipogenesis differentiation and other study is consistent with our study [27].

Next, we explored signal upstream of GSK-3β. Activation of ERK and AKT could affect activation status of GSK-3β [28]. During ADM treatment, the addition of D and DA did not affect the phosphorylation of AKT (S473) which can phosphorylate and inactivate GSK-3β. (Appendix A). Whereas, expression of phosphorylated ERK at Thr 202/Tyr 204 decreased after adipogenic induction and was restored by treatment of D and DA. Thus, these results suggest that GSK-3β phosphorylation at S9 might be regulated by phospho-ERK (T202/Y204) and is important for anti-adipogenic effect induced by D and DA in ASCs.

We then conducted experiments to see if inhibition of adipogenesis by D and DA is mediated through activation of β-catenin in ASCs. D and DA induced upregulation of C/EBPα and PPARγ was suppressed by si-β-catenin, suggesting that β-catenin signaling could play a critical role in the anti-adipogenic effect of D and DA (Figure 5). Our results suggest β-catenin signaling could inhibit adipogenesis through downregulation of C/EBPα and PPARγ expression. Recently, β-catenin has been shown to block interaction between the T-cell factor/lymphoid enhancer-binding factor (TCF/LEF)-binding domain of β-catenin and the catenin-binding domain of C/EBPα and PPARγ [8,9,10]. These studies, suggesting β-catenin as upstream regulator of C/EBPα - PPARγ pathways support our results. Altogether, inhibition of adipogenesis by D and DA could be mediated via β-catenin- C/EBPα - PPARγ signaling axis [27,29,30].

Also, microenvironmental factors of ASCs may affect the adipogenesis. Specially, inflammatory microenvironment from adipocytes have emerged as an important contributor to nutrient overload in adipocytes of adipose tissue [31]. In our results, we performed qRT-PCR after adipogenic induction with ADM treatment for 12 days to determine whether D and DA affect the expression of inflammatory transcription factors. The mRNA level of inflammatory genes, IL-6 and IL-1β, were significantly increased by ADM exposure. However, upregulated mRNA level of IL-6 and IL-1β with ADM treatment were significantly suppressed by addition of D and DA to ADM (Appendix A). Therefore, further studies are needed to evaluate the anti-obesity effect of D and DA on the interaction between ASCs and VAT microenvironment.

In summary, our study shows that D and DA inhibit adipocyte differentiation of ASCs from human VAT through the activation of GSK-3β/β-catenin signaling pathway (Figure 6).

Prevalence of visceral obesity has been gradually increasing in older people and especially in postmenopausal women [32,33]. Visceral obesity is a major contributor to cardiovascular disease which is the number one cause of death [17,34,35]. Several chemicals have been approved by FDA as anti-obesity drugs. Those drugs suppress appetite by amplifying satiety signals in brain or reducing nutrient absorption through inhibition of pancreatic lipase enzyme. However, current anti-obesity drugs have been reported to have severe adverse effects such as cardiovascular dysfunction, anxiety, and suicidal thoughts [36]. Therefore, there is an increasing demand for development of an anti-obesity drug that is both safe and specific for visceral obesity. In this context, D and DA might be used as natural anti-obesity agents for persons with visceral obesity.

## Figures and Tables

**Figure 1 nutrients-12-00013-f001:**
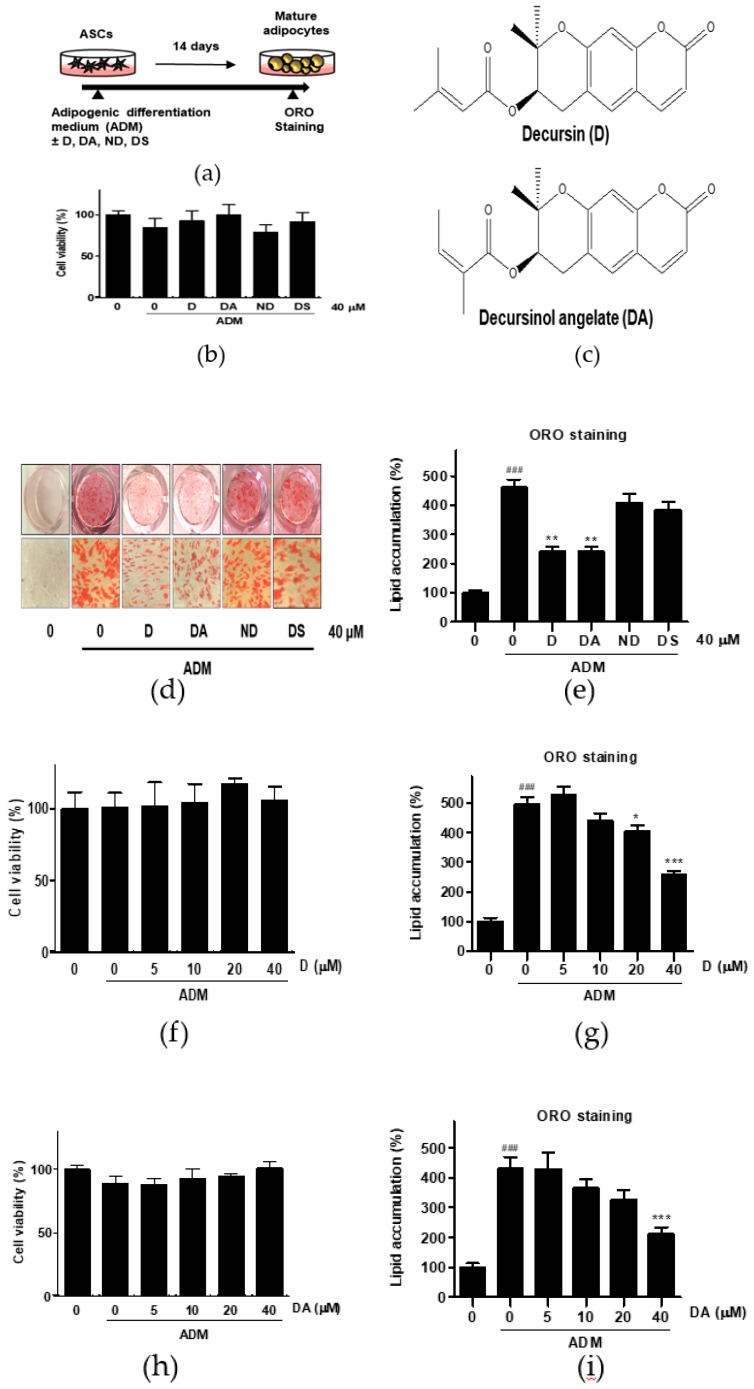
The effect of four different coumarin compounds from *Angelica gigas* Nakai (AGN) on adipogenesis of ASCs. (**a**) An experimental scheme. Post-confluent ASCs were cultured with or without the coumarin compounds (40 μM) on day 14 after ADM treatment. The cells were then stained using Oil Red O (ORO). (**b**) Chemical structure of decursin (D) and decursinol angelate (DA). (**c**) The effect of four coumarin compounds on cell viability. Cell viability was determined by MTT assay on day 2 after ADM treatment. (**d**) ORO staining of ASCs on day 14 after ADM treatment (Magnification: 200×). (**e**) The intracellular lipid content was quantified by extracting ORO stained lipid droplets with 100% isopropanol and optical density (OD) was measured at 495 nm. Values are expressed as mean ± SD of three independent experiments. ^###^
*p* < 0.001 vs. undifferentiated group, ** *p* < 0.01 vs. ADM treated group. (**f**, **h**) The effects of D and DA on cell viability at various concentrations (5, 10, 20, and 40 μM). Cell viability was determined by MTT assay on day 2 after ADM treatment. Values are expressed as mean ± SD of three independent experiments. (**g**, **i**) The intracellular lipid content after treatment with D and DA at various concentrations (5, 10, 20, and 40 μM). Values are expressed as mean ± SD of three independent experiments. ^###^
*p* < 0.001 vs. undifferentiated group, * *p* < 0.05 and *** *p* < 0.001 vs. ADM treated group. ADM: adipogenic differentiation medium.

**Figure 2 nutrients-12-00013-f002:**
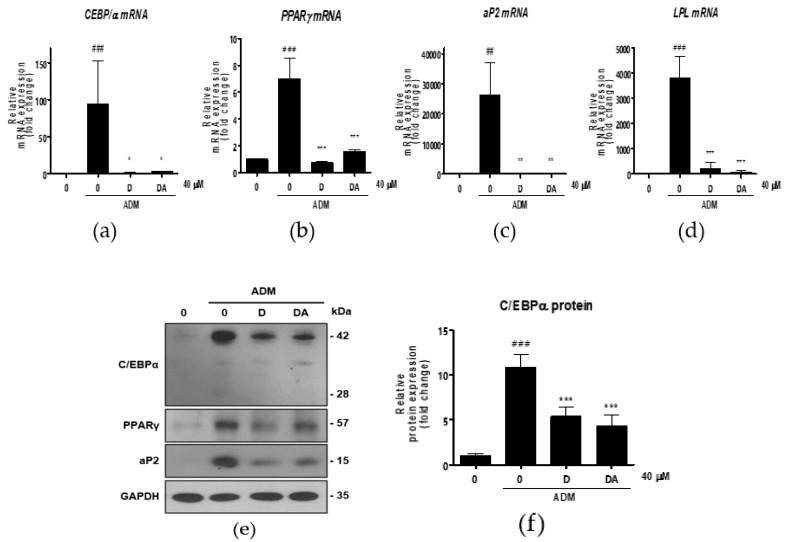
The effect of D and DA on the expression of adipogenic and lipogenic markers in ASCs. (**a**–**d**) The relative mRNA expression of adipogenic genes, C/EBPα, PPARγ, aP2, and LPL analyzed by qRT-PCR on day 3 after ADM treatment. Values are expressed as mean ± SD of three independent experiments. ^##^
*p* < 0.01 and ^###^
*p* < 0.001 vs. undifferentiated group, * *p* < 0.05, ** *p* < 0.01 and *** *p* < 0.001 vs. ADM treated group. (**e**–**h**) The protein expression of adipogenesis related proteins, C/EBPα, PPARγ and aP2, analyzed by Western blot on day 4 after ADM treatment. Values are expressed as mean ± SD of three independent experiments. ^###^
*p* < 0.001 vs. undifferentiated group, *** *p* < 0.001 vs. ADM treated group. (**i**–**k**) The relative mRNA expression of lipogenic genes, fatty acid synthase (FAS), acetyl CoA carboxylase (ACC), and SREBP-1c analyzed by qRT-PCR on day 12 after ADM treatment. Values are expressed as mean ± SD of three independent experiments. ^#^
*p* < 0.05 vs. undifferentiated group, * *p* < 0.05 vs. ADM treated group. (**l**–**n**) The protein expression of lipogenesis related proteins, FAS and ACC analyzed by Western blot on day 12 after ADM treatment. Values are expressed as mean ± SD of three independent experiments. ^###^
*p* < 0.001 vs. undifferentiated group, * *p* < 0.05 and ** *p* < 0.01 vs. ADM treated group. ADM: adipogenic differentiation medium.

**Figure 3 nutrients-12-00013-f003:**
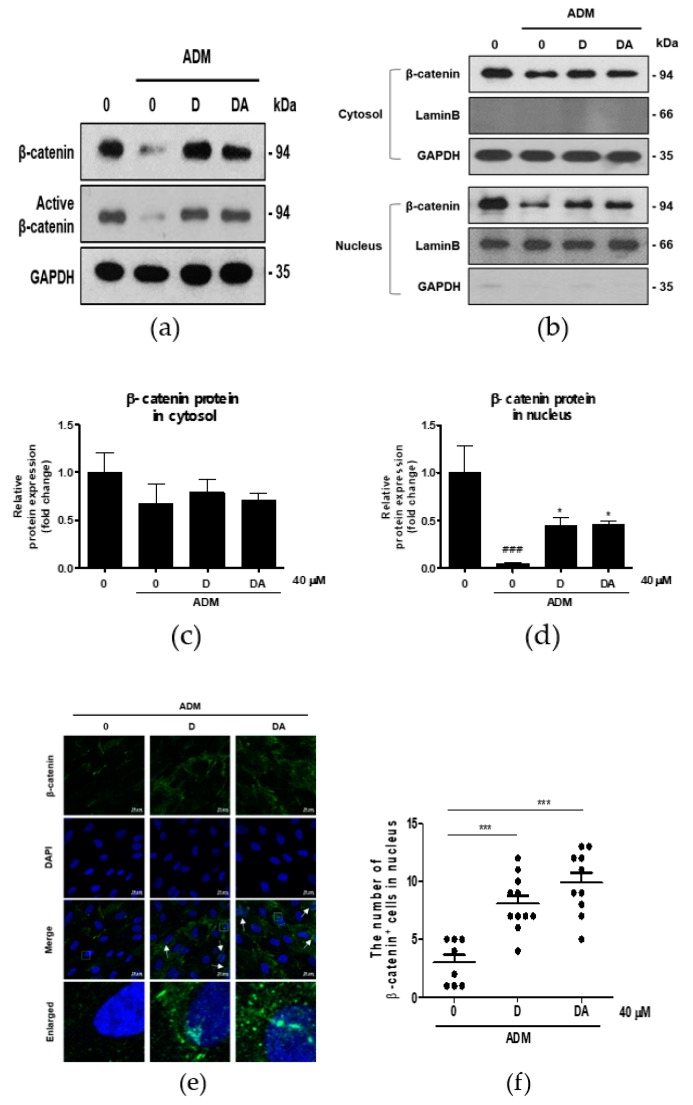
The effect of D and DA on β-catenin signaling in ASCs. (**a**) D and DA upregulated the expression of β-catenin and active β-catenin (non-phosphorylated β-catenin). ASCs differentiation into adipocytes was induced in the presence or absence of D and DA for 4 days and Western blot was performed. (**b**–**d**) Treatment with D and DA upregulated the expression of β-catenin in the nucleus during adipogenesis of ASC. After adipogenesis induction with or without D and DA for 4 days in ASCs, the expression of β-catenin in nuclear and cytoplasmic fraction was analyzed by Western blot. Values are expressed as mean ± SD of three independent experiments. ^###^
*p* < 0.001 vs. undifferentiated group, * *p* < 0.05 vs. ADM treated group. (**e**) The immunocytochemistry images by confocal microscope show localization of β-catenin relative to nucleus (DAPI stained) with D and DA treatment. Bar = 20 μm. (**f**) The dot plot represents the number of cells positive for β-catenin signal in nucleus. The dot plot represents two independent experiments, each group containing 90–100 cells counted from confocal microscopy images. *** *p* < 0.001 vs. ADM treated group. ADM: adipogenic differentiation medium.

**Figure 4 nutrients-12-00013-f004:**
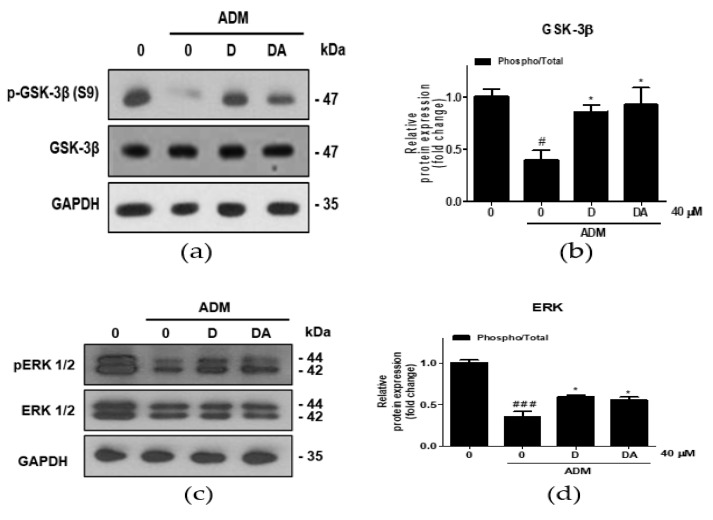
The effect of D and DA on inhibitory phosphorylation of GSK-3β, a major regulator of β-catenin stability in ASCs. (**a**,**b**) The protein expression of phosphorylated GSK-3β at S9 was analyzed by Western blot on day 4 after ADM treatment. Values are expressed as mean ± SD of three independent experiments. ^#^
*p* < 0.05 vs. undifferentiated group, * *p* < 0.05 vs. ADM treated group. (**c**,**d**) The protein expression of phosphorylated ERK 1/2 (T202/Y204), as an upstream kinase of GSK-3β analyzed by Western blot on day 4 after ADM treatment. Values are expressed as mean ± SD of three independent experiments. ^###^
*p* < 0.001 vs. undifferentiated group, * *p* < 0.05 vs. ADM treated group. ADM: adipogenic differentiation medium.

**Figure 5 nutrients-12-00013-f005:**
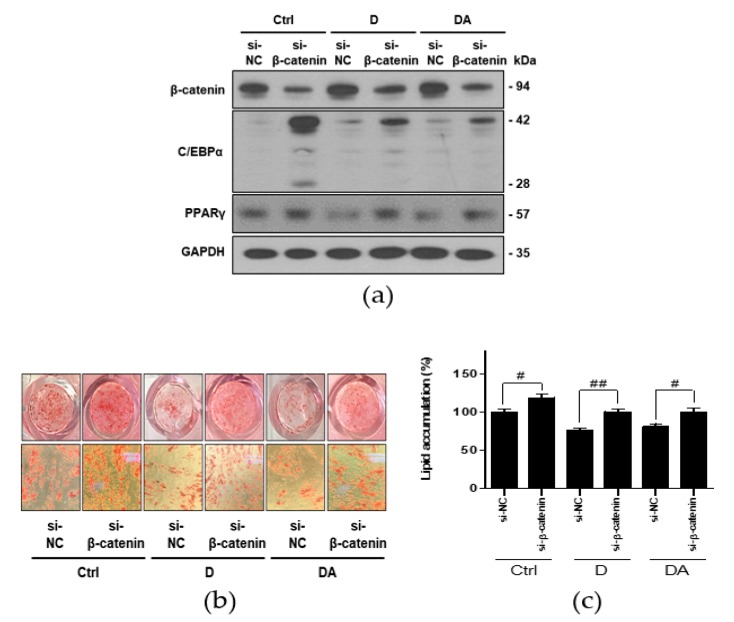
The effect of D and DA on ASCs after β-catenin knockdown. (**a**) The protein expression of β-catenin, C/EBPα, and PPARγ after silencing of β-catenin using siRNA in ASCs. ASCs were transfected with NC or β-catenin siRNA (50 nM) for 8 h and adipogenic induction was done for 4 days in the presence or absence of D and DA (40 μM). Western blot was performed on day 4 after ADM treatment. (**b**,**c**) β-catenin knockdown ameliorated inhibition of adipogenesis by D and DA in ASCs. After transfection of ASCs with si-β-catenin, adipogenesis was induced with or without D and DA (40 μM). ORO staining was performed on 14 days after ADM treatment. (Magnification: 200×) and ORO-stained lipid droplets were quantified. Values are expressed as mean ± SD of three independent experiments. ^#^
*p* < 0.05 vs. si-NC and ^##^
*p* < 0.01 vs. si-NC group. ADM: adipogenic differentiation medium; NC: negative control.

**Figure 6 nutrients-12-00013-f006:**
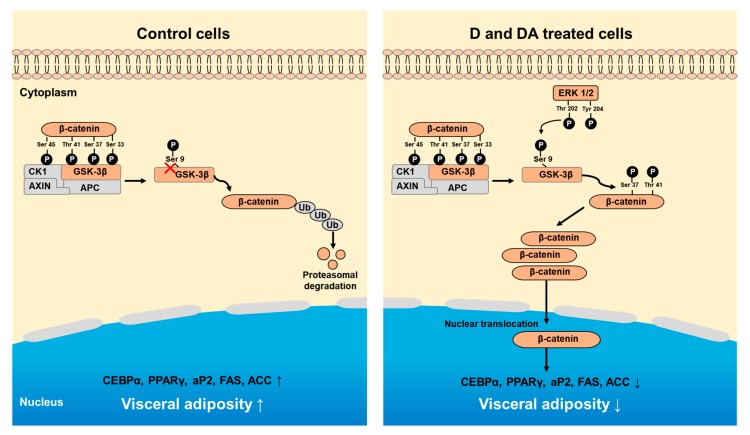
Schematic diagram showing D and DA on adipocyte differentiation through GSK-3β/β-catenin signaling in ASCs. Treatment of D and DA on ASCs increases the expression and nuclear translocation of β-catenin. Upregulation of β-catenin is mediated through inhibitory phosphorylation of GSK-3β (S9) by ERK. Overall, activated β-catenin by D and DA suppresses adipogenic markers, C/EBPα and PPARγ.

**Table 1 nutrients-12-00013-t001:** Clinical information of donors for adipose-derived stem cells (ASCs).

Donor No.	Sex	WHR	BMI (kg/m^2^)	Age (years old)
#49	Female	0.91	21.6	50
#50	Female	0.94	23.0	54
#71	Female	0.94	23.7	58
#73	Female	0.93	21.1	63
#91	Female	0.96	24.4	52

Human VAT samples were taken from intra-abdominal of donors (*n* = 5) undergoing gynecologic surgery. WHR, waist to hip ratio; BMI, body mass index.

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
