# Peer review of "Decursin and Decursinol Angelate Suppress Adipogenesis through Activation of β-catenin Signaling Pathway in Human Visceral Adipose-Derived Stem Cells"

_nutrients, 2019, doi:10.3390/nu12010013_

Round 1
Reviewer 1 Report
This is an interesting study showing the beneficial effects Decursin and decursinol on adipogenesis in adipose-derived stem cells isolated from visceral adipose tissue. The authors found that decursin and decursinol suppressed adipogenesis through activation of β-catenin signaling pathway. The study is technically sound, the main conclusion is supported by data but some issues should be addressed by the authors.
ORO staining is a semi quantitative approach for lipid determination and it quantifies neutral lipids (cholesterol ester, triglycerides). The authors should provide triglyceride and cholesterol measurements in cells.
Adipocyte differentiation could be also determined by SREBP-1c which regulates key lipogenic enzymes such as FAS ACC. Did the authors determine this nuclear receptor?
LPL (PPARy target) is a critical regulator of adipogenesis. The authors did not investigate this issue, neither discussed.
Reviewer 2 Report
The study presented by Park et al. describes the effects of coumarins present in Angelica gigas Nakai on adipogenesis in adipose-derived stem cells isolated from human visceral adipose tissue. It is well-written and very interesting, and the data presented are relevant and clearly shown. The authors found that two coumarins, decursin and decursinol angelate, were able to reduce the intracellular lipid content, in a dose-dependent manner, with not toxic effects on the cells, by downregulating adipogenic (CEBP/α, PPARγ, aP2) and lipogenic (FAS, ACC) markers. The authors propose herein that decursin and decursinol angelate promote ERK1/2 activation which, in turn, inhibits GSK-3β, and therefore allowing the translocation of β-catenin into the nucleus.
Minor comments:
1. Gene and protein names are shown in lowercase and capital letters indistinctly. For example, in Materials and Methods, 2.4. qRT-PCR, gapdh, cebpα, pparg, ap2, must be written according to guidelines (GAPDH, CEBP/α, PPARγ, aP2). In Figure 2 (legends and figure titles), the authors wrote cebpa, pparg, ap2, C/EBPα, among others throughout the manuscript. Please, review the whole manuscript and fix those inconsistencies.
2. Lines 157 and 169: cell density is expressed as x 106 cells per well. Number 6 must be a superscript.
3. Line 229: “while the increases in of adipogenic marker”, in or of?
4. ASC were treated with ADM for 14 days in Figure 1 and 12 days in Figure 2. Why? Did the authors make a time-course experiment to find the best time point? In such a case, please show or explain the reason.
Round 2
Reviewer 1 Report
The authors answered satisfactorily the concerns of this reviewer